# *Trichoderma*: An Eco-Friendly Source of Nanomaterials for Sustainable Agroecosystems

**DOI:** 10.3390/jof8040367

**Published:** 2022-04-02

**Authors:** Mousa A. Alghuthaymi, Kamel A. Abd-Elsalam, Hussien M. AboDalam, Farah K. Ahmed, Mythili Ravichandran, Anu Kalia, Mahendra Rai

**Affiliations:** 1Biology Department, Science and Humanities College, Shaqra University, Alquwayiyah 11726, Saudi Arabia; 2Plant Pathology Research Institute, Agricultural Research Center (ARC), 9-Gamaa St., Giza 12619, Egypt; kamel.abdelsalam@arc.sci.eg; 3Plant Pathology Department, Faculty of Agriculture, Cairo University, Giza 12613, Egypt; hussien.abodlam@gmail.com; 4Biotechnology English Program, Faculty of Agriculture, Cairo University, Giza 12613, Egypt; farahkamel777@gmail.com; 5Department of Microbiology, Vivekanandha Arts and Science College for Women, Sankari 637303, Tamil Nadu, India; ms.microhoney@gmail.com; 6Electron Microscopy and Nanoscience Laboratory, Punjab Agricultural University, Ludhiana 141004, Punjab, India; kaliaanu@pau.edu; 7Department of Microbiology, Nicolaus Copernicus University, Lwowska 1, 87100 Torun, Poland; mahendra.rai@v.umk.pl

**Keywords:** beneficial microbes, biocontrol agents, *Trichoderma*, *Hypocrea*, nanostructures

## Abstract

Traditional nanoparticle (NP) synthesis methods are expensive and generate hazardous products. It is essential to limit the risk of toxicity in the environment from the chemicals as high temperature and pressure is employed in chemical and physical procedures. One of the green strategies used for sustainable manufacturing is microbial nanoparticle synthesis, which connects microbiology with nanotechnology. Employing biocontrol agents *Trichoderma* and *Hypocrea* (Teleomorphs), an ecofriendly and rapid technique of nanoparticle biosynthesis has been reported in several studies which may potentially overcome the constraints of the chemical and physical methods of nanoparticle biosynthesis. The emphasis of this review is on the mycosynthesis of several metal nanoparticles from *Trichoderma* species for use in agri-food applications. The fungal-cell or cell-extract-derived NPs (mycogenic NPs) can be applied as nanofertilizers, nanofungicides, plant growth stimulators, nano-coatings, and so on. Further, *Trichoderma*-mediated NPs have also been utilized in environmental remediation approaches such as pollutant removal and the detection of pollutants, including heavy metals contaminants. The plausible benefits and pitfalls associated with the development of useful products and approaches to trichogenic NPs are also discussed.

## 1. Introduction

The use of myco-nanoparticles in agriculture is still in its early phases of research, especially in terms of their interactions with agriculturally beneficial microorganisms. A few in vitro experiments have been published, while in vivo investigations are currently taking place in greenhouse conditions [1,2,3]. Microorganisms, often known as nanofactories or nanoparticle producers, have potential because their cellular machinery may be altered to make the synthesis of NPs easier [4,5]. The biogenic synthesis based on fungi has several benefits in terms of efficiency and the generation of diverse metabolites under optimal circumstances. Furthermore, because fungi are natural producers of a wide range of antimicrobial compounds, using them as a capping agent of nanoparticles might result in a synergistic antimicrobial impact with metal NPs against pathogenic microorganisms [6].

Among the microbial agents, significant diversity of intracellular and extracellular proteins and enzymes that function as reducing agents makes fungal bioagents more suited than bacteria [7]. *Trichoderma* strains have a long history of effectiveness as biocontrol agents against a variety of pathogenic microorganisms. Furthermore, recent research has demonstrated that these fungi improve plant resilience, growth, and development, resulting in increased yield output [8,9]. One of the strategies for decreasing the harmful effects of heavy metals on plants is to use *Trichoderma harzianum.* Several *Trichoderma* species have been identified for the synthesis of a variety of important secondary metabolites such as plant growth regulators, antibiotics, and enzymes, which are mostly utilized to defend plants against pathogens [9]. Additionally, the metabolites produced and secreted by *Trichoderma* species in the culture filtrates are known to possess antimicrobial, anticancer, and antioxidant properties [10].

Recently, the use of agrochemicals incorporating nanostructured materials has emerged as a viable agricultural option [11]. Several *Trichoderma* species have been employed in nanotechnology, primarily for the synthesis of metal nanoparticles (Figure 1). Their resistance to many nano compounds has recently been discovered, but little is known about their contribution to the production of metallic NPs via tolerance to these chemicals and how these aspects affect *Trichoderma* relationships. Enzymes such as reductases, which can operate as bioreductive agents in the biofabrication of NPs, can be considered the key biocomponents for the *Trichoderma*-mediated mycosynthesis of NPs [12]. *Trichoderma* is an easy-to-manage fungus with several physiological and technological advantages [13]. The biosynthesis of metal nanoparticles using the advantageous *Trichoderma* hyphal extracts is a straightforward, environmentally friendly, and cost-effective method. Secondary metabolites released by *T. harzianum* operate as capping and reducing agents, which contribute to the biological activity and green synthesis of AgNPs using *Trichoderma* [14]. It is a quick, economically feasible, environmentally safe, nontoxic metal nanoparticle synthesis technique well-suited for large-scale production [15]. As a result, utilization of this green bio-based, environmentally friendly, and economically feasible approach can be a plausible alternative solution for the successful manufacturing of sustainable nanomaterials. In this review, the information on various uses of *Trichoderma* genus to develop techniques for the mycosynthesis of metal NPs and their uses in agroecosystems is presented. Further, the probable role of novel *Trichoderma*-NP bioconjugates as a viable alternative for sustainable agriculture will be identified.

## 2. Beneficial Effects of *Trichoderma* in Agroecosystems

The *Trichoderma* genus includes fungal species that are economically significant because of their plant growth and performance-promoting actions, such as enhanced nutrient availability, the mycoparasitism of plant pathogens, and the priming of plant defense. The biodiversity of the genus *Trichoderma* provides fresh insight into their uses, particularly in agriculture and industry, as well as their potential use as biofungicides and biofertilizers in the field. *Trichoderma* spp. boost plant development, initiate plant defense, and aid plant growth in response to a variety of biotic and abiotic challenges, such as soil dryness, excessive salt content, and the presence of poisonous metal ions [16]. *Trichoderma* spp. are well-known plant growth-promoting fungi (PGPFs) with the potential to compete with pathogenic microorganisms while also promoting plant health [17]. *Trichoderma* species are extremely significant as biocontrol agents against a variety of plant pathogenic fungi and, hence, serve as viable alternatives to synthetic fungicides. *Trichoderma* species have been used as biological control agents for the management of plant pathogenic microbes, including fungi and bacteria [18]. *Trichoderma* and their secondary metabolites released into the rhizosphere may have an impact on plant development and nutrition, as well as the induction of systemic resistance and the biocontrol of pathogenic bacteria [18]. Plants and *Trichoderma* communicate bidirectionally via numerous signal molecules, resulting in a beneficial symbiotic relationship [16]. It can colonize plant roots by sensing nutrients secreted from the roots in the rhizosphere; interact with plants by producing various MAMP (microbe-associated molecular pattern) and damage-associated molecular pattern (DAMP) molecules; stimulate an immune response, suppressing many plant pathogens through the use of systemic acquired resistance (SAR) and induced systemic resistance (ISR) mechanisms; and act as a systemic resistance inducer in the plant [16,17]. Colonization of roots and leaves by *Trichoderma* can prime plant defense, allowing for powerful plant responses to following pathogen threats [19]. However, in addition to biopesticide action, certain *Trichoderma* strains have been shown to exhibit biostimulant activity, plant growth promotion, enhanced yield and nutritional quality, and the ability to mitigate the negative effects of abiotic stressors [20,21,22]. Therefore, *Trichoderma* has evolved from a mycoparasitic biocontrol agent (BCA) to one with numerous features such as pathogen antagonism, pathogen competition for resources, induction of systemic resistance in the host, overall plant growth promotion, and reduction of abiotic stressors. Furthermore, while being previously described primarily as soil and root colonizers, it is now clear that numerous *Trichoderma* species are endophytic [23]. As a result, it is not surprising that *Trichoderma* is found to be an effective beneficial biological agent, with active ingredients in over 200 agricultural products such as biopesticides, biofertilizers, bio-growth enhancers, and bio-stimulants that are marketed across the globe [15,24,25].

Beneficial fungi, including the *Trichoderma* species, are integral components of the decomposer microflora population, which plays a vital role in bioconcentration, decontamination, and even degradation/removal of the xenobiotics added to the ecosystem due to intentional and/or unintentional incorporation of a variety of contaminants. This removal or decontamination is referred to as ‘bioremediation’ and involves the use of microbial enzymes to convert toxic metal compounds into nonhazardous chemicals [26]. In a research report, *T. viride* exhibited the ability to break down and use nitrogenous (trinitrotoluene, TNT) explosives at doses of 50 and 100 ppm as the N-source to meet the nitrogen requirements for normal development [27]. Besides nitrogenous explosives, the potential hydrocarbon-degrading abilities of *Trichoderma* species can be useful to bioremediate diesel oil spills in an aquatic ecosystem, ensuring the protection of the environment [28]. Another report reiterated the use of *Trichoderma* species, particularly *T. harzianum* strain T22, for ensuring the biodegradation of diesel fuel, allowing it to be used as a carbon source [29]. Further, the physiological and metabolic versatility of various *Trichoderma* fungal species can be utilized for the remediation of heavy metals from different eco-niches. *T. lixii* CR700 demonstrated excellent Cu removal capability across a wide pH range. In the removal of Cu, *T. lixii* CR700 employs simultaneous surface sorption and accumulation processes [30].

Different *Trichoderma* species are, thus, effective natural decomposition agents to speed up the degradation of organic materials. The benefit will be two-fold as the nutrients released from the xenobiotic will be available in the plant rhizosphere and can be utilized and taken up by the plant [9]. The presence of *Trichoderma* with alfalfa seedlings boosted the soil’s N, P, and K content and alfalfa biomass [31]. The conjugate use of the *Trichoderma* and chemical fertilizer improved the plant’s nutritional quality, productivity, and vegetative and reproductive development. This can cut agricultural expenses while also reducing pollutants [32]. Overall, *Trichoderma* acts as a nutrient mobilizer, improving the quality and yield components of the crops.

Many studies have indicated that *Trichoderma* spp. are particularly efficient at degrading pesticides and can be useful in an integrated pest control strategy as well as for reducing pesticide residual effects. *Trichoderma* spp. have been reported to degrade benzimidazole fungicide (Carbendazim) [33], chlorpyrifos [34], penthiopyrad [35], and 2,2-dichlorovinyl dimethyl phosphate (DDVP) [36]. Wu et al. [37] proposed the tolerance mechanism of *T. asperellum* TJ01 to dichlorvos. *Trichoderma* spp. produce a variety of volatile organic compounds (VOCs) possessing various chemical properties. These metabolites are critical in agricultural, food, and pharmaceutical applications [38]. *T. reesei* has not been proven to be hazardous to humans [39]. The biosynthesis of silver nanoparticles by the fungus *T. reesei* is preferable in terms of safety, economics, and large-scale production capability [39]. As a result, the use of *Trichoderma* should be encouraged since it offers sustainable agriculture by minimizing the use of hazardous chemicals in agriculture (Figure 2).

## 3. Biosynthesis of Nanoparticles by *Trichoderma* Genus

Fungal nanotechnology is one of the most popular options because of the vast range of benefits it has over bacteria, actinomycetes, plants, and other organisms in terms of physicochemical qualities [40]. When it comes to the biological generation of NPs, fungi outperform the majority of microorganisms in terms of efficiency. This is due to the ability of the fungi to produce a large variety of bioactive metabolites and metal accumulation properties and have improved processes, all of which are beneficial [41]. Because of their capacity to tolerate metals and accumulate metals in their tissues, fungi have emerged as an important branch for the biosynthesis of nanoparticles [42]. Green synthesis of nanoparticles is nontoxic as it involves the utilization of safe reagents, which makes it more cost-effective and eco-friendlier compared to the traditional methods [14]. Fungi offer several other benefits, such as simplicity of management and cultivation, no requirement of complicated components, production of a large amount of biomass and metabolites, high cell wall-binding capability, and ability to absorb large amounts of metal [43]. The next section describes the production of myconanoparticles from several *Trichoderma* species.

### 3.1. Silver Nanoparticles

The first report on the biosynthesis of silver nanoparticles with regulated properties from a nonpathogenic and economically viable biocontrol agent involved the incubation of the cell-free filtrate of *T. asperellum* for 5 days at 25 °C containing AgNO_3_ (1 mM) [44]. The use of UV–vis spectroscopy revealed the kinetics of the reaction with a strong surface plasmon resonance band observed at 410 nm, indicating the synthesis of silver nanoparticles. Based on the results of TEM and XRD studies, the size of the silver nanoparticles ranged from 13 to 18 nm [44]. *T. reesei* has also been used for the extracellular biosynthesis of AgNPs for the first time [39], allowing for the synthesis of AgNPs on an industrial scale. Fluorescence emission spectroscopy was used to produce detailed information on the progress of the decrease of silver nitrate (formation of silver nanoparticles) on the nanosecond timescale. The quantitative analysis of the reaction products was carried out using Fourier transform infrared spectroscopy (FTIR). The TEM images revealed the morphologically varied forms and crystalline nature of the AgNPs, with diameters ranging from 5 to 50 nm. Likewise, another study showcased *T. virens* to be the most efficient producer of AgNPs when compared to a total of 75 isolates from five distinct *Trichoderma* species. Every 24 h, the highest plasmon band was recorded at 420 nm, reaching maximum intensity at 120 h. The morphology of NPs was validated by high-resolution transmission electron microscopy (HRTEM), which revealed that nanoparticles were single or aggregated, spherical and homogeneous in shape, and ranged in size from 8 to 60 nm [45].

*T. harzianum* cell filtrate was used to synthesize AgNPs in a simple green and eco-friendly manner, without the need for any toxic reducing agents, capping agents, or dispersion agents. Temperature and AgNO_3_ concentration both had a considerable impact on AgNP production, as evidenced by UV–vis spectra. AgNPs were shown to be stable for three months, as confirmed through the DLS study. The synthesized AgNPs possessed face-centered cubic symmetry with a size range of 10–51 nm [46]. *T. harzianum* extracellular filtrate is an entomopathogenic fungus used for AgNP synthesis with distinct characteristics. The reduction rate of silver ions was assessed using a UV-visible spectrophotometer, which revealed a maximum absorption at 433.5 nm. Fourier transform infrared spectroscopy identified specific fungal metabolite functional groups responsible for the synthesis of AgNPs. The linkage between amino acid residues was shown by the amide group through the vibration of the N–H bend. X-ray diffraction analysis confirmed that the mycosynthesized AgNPs were oval in shape, crystalline in nature, and monodispersed in colloidal form, as well as exhibiting high purity. The surface properties of the generated AgNPs, as depicted by scanning electron microscopy, involved no direct contact even inside aggregates, indicating that AgNPs have been stabilized by a capping agent, and the size was around 10–20 nm [47]. Elgorban et al. [48] studied the synthesis of biogenic silver nanoparticles at room temperature in darkness using the fungus *T. viride*. Centrifugation was used to increase the concentration of AgNPs. The bioreduction of silver nanoparticles (AgNPs) was seen spectrophotometrically, and the AgNPs under investigation were characterized using UV–vis, TEM, and SEM. *T. viride* AgNPs were reported to be stable and polydispersed globular particles, with diameters ranging from 1 to 50 nm. They can be used as a reducing agent in the biogenic production of silver nanoparticles. *T. harzianum*-derived AgNPs possessed a spherical shape, acceptable polydispersity, and a size distribution between 20 and 30 nm, with no aggregates [4]. Biological silver nanoparticles were generated extracellularly utilizing the fungus *T. longibrachiatum*, with the fungal cell filtrate employed as a reducing and stabilizing agent in the nanoparticle manufacturing process. Incubation at 28 °C for 72 h with fungal biomass without agitation resulted in AgNP biosynthesis. The production of spherical nanoparticles with sizes ranging from 5 to 25 nm was observed by TEM [49]. Saravanakumar and Wang [50] reported, for the first time, the synthesis of anisotropic structured AgNPs using *T. atroviride* and investigated their biomedical properties. The observation of plasmon resonance at 390–400 nm in the UV–vis spectrum verified the formation of AgNPs. FTIR, transmission electron microscopy, and EDX examination revealed a significant percentage signal of anisotropic structural AgNPs, with sizes ranging from 15 to 25 nm. The *Trichoderma* filtrate contained biochemicals capable of bio-reducing silver nitrate (AgNO_3_), which are spherical NPs and nontoxic at low concentrations [51].

Extracellular biosynthesis of AgNPs from AgNO_3_ solution using *T. reesei* PF biomass cell-free water extract (CFE) has been described [52]. The optimal generation of AgNPs was achieved when *T. reesei* fungi were cultured in conditions containing 0.1 percent corn steep liquor, 10% biomass extracted to produce CFE, and 10 mM AgNO_3_. HRTEM micrographs verified the presence of Ag metallic nanoparticles in the crystal phase. The TEM pictures revealed the development of AgNPs clusters with sizes ranging from 1–4 to 15–25 nm. The NPs were stabilized by the capping action of the biomolecules, as indicated by FTIR spectroscopy. The zeta-potential of the prepared AgNPs was negative [52]. AgNPs were effectively synthesized using *T. harzianum* filtrates cultured in the presence and absence of enzymatic stimulation of the *S. sclerotiorum* cell wall, resulting in nanoparticles with distinct physicochemical properties. The DLS approach was utilized to determine hydrodynamic diameters, yielding mean hydrodynamic diameters of 57.02 ± 1.75 nm for AgNP-TS and 81.84 ± 0.67 nm for AgNP-T, respectively. The NTA approach yielded nanoparticle sizes and concentrations that differed in AgNP-TS and AgNP-T particle sizes of 88.0 ± 7.3 and 182.5 ± 6.9 nm, respectively [43]. *T. longibrachiatum* DSMZ 16,517 mycelial cell-free filtrate (MCFF) bioreduced silver ions (Ag+) to their metallic nanoparticle state (Ag^0^), as shown by AgNPs. The DLS examination revealed average AgNP size and zeta potential values of 17.75 nm and 26.8 mV, suggesting the stability of the synthesized AgNPs. The crystallinity of the mycosynthesized AgNPs, with an average size of 61 nm, was confirmed by the XRD pattern. The FESEM and HRTEM images revealed non-agglomerated spherical, triangular, and cuboid AgNPs, with sizes ranging from 5 to 11 nm. The FTIR analysis of the mycosynthesized AgNPs confirmed MCFF’s activity as a reducing and capping agent [53]. *T. atroviride*-hosted *Chiliadenus montanus* was shown to be the best candidate for the synthesis of mycogenic AgNPs among all investigated species. HRTEM was used to describe these AgNPs, which revealed a dispersion of spherical AgNPs ranging in size from 10 to 15 nm. By using the agar well diffusion technique, mycosynthesized AgNPs were compared to chemically synthesized AgNPs for their antibacterial, anticandidal, and antifungal activities over six pathogenic bacteria and four pathogenic fungi in vitro [54]. Chitin-induced exo-metabolites extracted from the varied and stress-tolerant *T. fusant* Fu21 were used to synthesize green AgNPs with sizes ranging from 59.66 to 4.18 nm (SEM), the spherical shape of nanoparticles with proven purity, stability (51.2 mV zeta potential), and nanoparticle polydispersity [55].

Amazon fungus *Trichoderma* sp. strain (TCH 01) was isolated from *Bertholletia excelsa* (Brazil-nut) seeds, and the soil was identified to biosynthesize the AgNP, as demonstrated by the RPSL band and UV–vis spectroscopy analysis. According to the TEM picture, the particles were polydispersed and spherical in form. AgNPs varied in size from 14 to 25 nm. All AgNPs exhibited incipient instability for the zeta potential. When grown at pH 5 for 9 days, smaller nanoparticles and higher polydispersity indexes were obtained [56]. Among 15 isolates of the *T.* species tested for the manufacture of AgNPs, a cell-free aqueous filtrate of *T. virens* HZA14 generating gliotoxin produced the best yield for the production of AgNPs. Electron microscopy tests revealed that AgNPs were 5–50 nm in size and had spherical and oval forms with smooth surfaces [57]. UV–visible spectrophotometry, FTIR, EDS, DLS, XRD, and SEM were used to analyze the AgNPs produced from *T. harzianum* culture filtrate. The surface plasmon resonance of synthesized particles created a peak with a center wavelength of 438 nm. According to the DLS research, the average size of AgNPs is 21.49 nm. SEM was used to determine the average size of AgNPs, which was 72 nm. The cubic crystal structure determined by XRD analysis validated the particles’ identification as silver nanoparticles [15]. As an alternative to traditional chemical and physical techniques, a green method for the synthesis of AgNPs has been provided. *T. reesei* fungus biomass was exploited as a green and renewable source of reductase enzymes and metabolites capable of converting Ag^+^ ions into AgNPs. *T. reesei* is an appropriate reagent for the production of monodisperse AgNPs that are stabilized by the capping effect of biomolecules. Trichogenic synthesis of AgNPs mediated by the *Trichoderma* genus is shown in Figure 3.

### 3.2. Zinc Oxide Nanoparticles

*T. harzianum* (PGT4), *T. reesei* (PGT5), and *T. reesei* (PGT13) monocultures and co-cultures (PGT4 + PGT5 + PGT13) generated secondary metabolites useful in the mycosynthesis of zinc oxide nanoparticles (ZnO NPs). *Trichoderma*—D-glucanzinc oxide nanoparticles (T—D-glu-ZnO NPs) were synthesized under optimal conditions using the fungal mycelial water extract (FWME) obtained from *T. harzianum*. The effective conjugation of D-glucan from barley with T-ZnONPs was confirmed by PACE and FTIR. The optimized T—D-glu-ZnO NPs had a spherical form with a mean size of 30.34 nm. T—D-glu-ZnONPs greatly suppressed the development of *Staphylococcus aureus* within roundworms while also enhancing roundworm growth [58]. Co-cultivation can induce the production of new secondary metabolites more effectively than monocultures. Another study involving the biosynthesis of ZnONPs showed the formation of NPs with crystalline structures free of impurities, according to PXRD analysis. The size of the crystalline particles ranged between 12 and 35 nm. These biosynthesized ZnONPs showed antibacterial efficacy against the rice cause of Bacterial Leaf Blight, *Xanthomonas oryzae* pv. *oryzae* [59]. *T. harzianum* is a possible fungal antagonist that is employed in the extracellular manufacture of ZnONPs. TEM imaging revealed that ZnONPs’ crystalline structure comprises hexagonal, spherical, and rod-shaped particles in a mixture of exceptionally small particles. The size range of the generated ZnONPs was 8–23 nm [3].

### 3.3. Copper Nanoparticles

*T. asperellum* cell-free extract was used to create copper oxide nanoparticles (TA-CuONPs). TA-CuO NPs were found to be crystalline with spherical particles. The CuONPs ranged in size from 10 to 190 nm, with an average diameter of 110 nm [60]. Consolo et al. [61] revealed, for the first time, the numerous extracellular biosynthesis of NPs from *T. harzianum*, as well as the synthesis of CuO and ZnONPs from this fungus. Biogenically synthesized Ag, CuO, and ZnO nanoparticles were made by employing a cell filtrate of a strain of *T. harzianum* as a reducer and stabilizer agent. When the ZnO NPs were evaluated against different target microorganisms, the potential of Ag and CuO for phytopathogen control was emphasized. Biocontrol agents *Pseudomonas fluorescens*, *T. atroviride,* and *Streptomyces griseus* were used in the production of copper and silica nanoparticles. *T. harzianum* was effectively used in the synthesis of copper nanoparticles. CuNPs were tested for antibacterial properties against two bacteria, *Staphylococcus aureus* and *Escherichia coli*, using a simple green and eco-friendly method [62]. UV–vis spectrometry, TEM, and EDAX analysis were used to analyze silica and copper nanoparticles. Nanoparticles were discovered to be aggregated and irregularly spherical. The size of silica nanoparticles varied from 12 to 22 nm, whereas the size of copper nanoparticles ranged from 5 to 25 nm [63].

### 3.4. Selenium Nanoparticles

The activity of biosynthesized selenium nanoparticles (SeNPs) using *T. asperellum* culture filtrate against *Sclerospora graminicola*, the cause of mildew disease in pearl millet, was higher. SeNPs were found to be hexagonal, near-spherical, and irregular in form, with sizes ranging from 49.5 to 312.5 nm. The size of SeNPs was inversely related to their biological activity [64]. Selenium nanoparticles (TSNPs) biosynthesized from *T. harzianum* JF309 were compared to conventional SNPs. The liquid metabolites of eight *Trichoderma* strains were collected, and modified biogenic synthesis was achieved. SNPs were found to be spherical or pseudo-spherical in shape, but TSNPs were more irregular. TSNPs were discovered to be somewhat larger than standard SNPs. The antifungal impact of TSNPs was far superior to that of standard SNPs [65]. The *Trichoderma* sp. WL-Go culture broth was used to create simple and cost-effective selenium nanoparticles. SeNPs had a partial size range of 20–220 nm, with an average diameter of 147.1 nm [66].

### 3.5. Other Nanoparticles

Gold nanoparticles were synthesized through a very fast and environment-friendly method within 10 min; this was the first time for such rapid biosynthesis of growth-promoting and plant pathogen control AuNPs. *T. viride* cell-free extract was treated with HAuCl_4_ at 30 °C for 10 min. TEM analysis confirmed that the NPs were widely dispersed and scattered in nature, with the bulk of them being spherical in form. The crystalline nature was verified by the SAED pattern. The size of the AuNPs ranged from 20 to 30 nm [67]. Chitosan nanoparticles were biogenically synthesized from *T. viride* and described using UV–vis spectroscopy, with FTIR verifying the functional groups of chitosan nanoparticles as OH, N–H, C–H, C=O, C–O, C–N, and P=O and electron microscopy demonstrating the roughly spherical form. DLS analysis determined the average size of chitosan nanoparticles to be 89.03 nm [68]. The *T. harzianum* (MF780864) isolate demonstrated the extracellular production of SiO2NPs from rice husks. UV, FTIR, DLS, and TEM were used to describe SiO_2_ NPs, which had a size of around 89 nm and shapes of oval, rod, and cubical particles [69].

The possibility of synthesizing FeNPs from *Trichoderma* species using simple methods has also been investigated. The presence of alkene, carboxyl, and phenol groups showed that the NPs were capped by the organism following the redox event. The generation of FeNPs from fungi such as *Trichoderma* species, which has the potential to generate more FeNPs than other bioresources, was proven [70]. *T. harzianum* is a biocontrol agent that is employed in the green manufacture of biogenic iron oxide nanoparticles for stabilization. Dynamic light scattering, nanoparticle tracking analysis, scanning electron microscopy, X-ray diffraction, and Fourier transform infrared spectroscopy techniques were used to assess the physicochemical properties of nanoparticles, and the results showed that the average size diameter of hematite (Fe_2_O_3_) nanoparticles are 207.2 nm [71]. Table 1 presents many metal NPs produced by different *Trichoderma* species, including AgNPs, ZnONPs, CuNPs and CuONPs, SeNPs, and other NPs, each with unique features and antimicrobial potential.

## 4. Production of Metal NPs by *Hypocrea*

In pure culture, cultures obtained from *Hypocrea lixii* ascospores developed the morphological species *T. harzianum. T. harzianum* is known as a cosmopolitan, ubiquitous species associated with a wide variety of substrates [72]. The ability of the fungus *Hypocrea lixii* to ingest and reduce copper ions to copper NPs was investigated. This approach verified the existence of proteins as stabilizing and capping agents around the copper nanoparticles. These results showed that the dead biomass of *H. lixii* is an economically and technically feasible solution for wastewater bioremediation and a possible candidate for industrial-scale synthesis of copper NPs [73]. Rapid synthesis of gold nanoparticles is possible in less than a minute using cell-free extracts of *T. viride* and recombinant *H. lixii* at various reaction temperatures. Another study was the first report of *T*. sp. involved in the rapid synthesis of gold nanoparticles that may function as an antimicrobial agent as well as an effective biocatalyst [67]. The fungus *H. lixii’s* dead biomass was discovered to be an effective instrument for the extracellular and intracellular synthesis of nickel oxide NPs as well as the absorption of hazardous metal ions from an aqueous solution. The fungus’s dead biomass played a significant role in the formation of metallic NPs, which are easily oxidized to nickel oxide in the media. The dead biomass also functioned as a stabilizer during the creation of nickel oxide NPs and so might be employed for nickel ion uptake during bioremediation procedures [74]. In addition, a new species, *H. virens*, has been discovered to be a teleomorph of *T. virens*, a species commonly used in biological control applications. *H. virens*, for example, is a possible biocontrol agent agonist of the *Ceratocystis paradoxa* pathogen that causes pineapple disease on sugarcane. The fungus *H. virens* was tested for its ability to synthesize AgNPs. A *H. virens* fungal biomass of extracellular and intracellular extract to aqueous silver nitrate solution for 72 h revealed that the appearance of a dark-brown color in the fungal biomass after reaction with Ag^+^ ions was a clear indicator of metal ion reduction and the formation of AgNPs by the fungal biomass [45].

## 5. Toxicity

In a recent study, the blood kinetics and tissue distribution of 20, 80 and 110 nm AgNPs were studied in rats [75]. The silver nanoparticles disappeared quickly from the blood and were distributed to all animal organs, independent of the NP size [75]. As a result, it can be identified that the AgNPs may have different toxicity and, hence, be associated with a distinct health risk. After repeated intravenous injections of silver nanoparticles, accumulation was observed [76]. The toxicity of AgNPs was determined by the dose and particle size [50]. In terms of toxicity, comparisons of cell lines 3T3 (mouse embryo fibroblasts), HeLa (human cervical adenocarcinoma), HaCaT (human keratinocytes), V79 (Chinese hamster pulmonary fibroblasts), and A549 (human epithelial adenocarcinoma) with controls revealed that the biogenic nanoparticles exhibited cytotoxic and genotoxic effects that varied depending on the cell line used and the exposure concentration. An interesting observation was that in the majority of these assays, the effects increased in direct proportion to the doses employed and were most strong at concentrations higher than those used in *S. sclerotiorum* inhibition evaluations. Concerning the impacts of nanoparticles in soil, there was evidence of some partial effects on bacteria, with probable microbiota recovery over time. Because the AgNP-T nanoparticles had no detrimental impacts on soybean germination and development, it is possible to conclude that this approach constitutes the first step towards the potential control of white mold in soybean crops using nanotechnology [4]. *T. atroviride*-induced AgNPs triggered cell changes, inhibition, or damage, as well as cytoplasmic compression in a concentration-dependent way. AgNPs significantly reduced cell viability with an inhibitory dose of 16.5 g/mL against human breast cancer cells (MDA-MB-231). Cell viability increased significantly with drug dosage concentration (NPs) against IMR 90, U251, and A549 lung cells, and human breast cancer cells MCF-7 and MDA-MB-231 [50]. The cytotoxicity and genotoxicity of AgNP-TS and AgNP-T nanoparticles were shown to be low in V79, 3T3, and HaCat cell lines. Guilger-Casagrande and colleagues [43] used, for the first time, photothermolysis to confirm the anticancer activity of biogenic CuO NPs. The TA-CuO NPs caused the photothermolysis of A549 cancer cells through ROS production, nucleus damage, mitochondrial membrane potential (m), and regulatory protein expression [60]. When SNPs and TSNPs were tested for cytotoxic action using human viability and mortality, neither SNPs nor TSNPs demonstrated hypertoxicity or lethality for the three cells [65]. At a concentration of 200 g ml^−1^, it possessed multi-mode antimicrobial action and has been determined to be nontoxic to humans. The lowest inhibitory concentration of green nanoformulations for the greater degrading activity of the pathogen fungal mycelium was determined to be 20 g Ag/mL green nanoformulations [55]. T—D-glu-ZnO NPs were not hazardous to NIH3T3 cells but had a dose-dependent inhibitory impact on human pulmonary carcinoma A549 cells, according to a cytotoxicity investigation. T-ZnO NPs and T—D-glu-ZnO NPs caused cancer cell death by necrosis and apoptosis, respectively [58]. The zebrafish model was used to assess the toxicity; there was no toxicity of nanocopper and nanosilica when exposed to 0.5, 3, and 30 g concentrations in respect of embryo viability, hatching rate, body mass index, and heartbeat counts [63]. The cytotoxicity of biogenic iron oxide nanoparticles was evaluated using several cell lines and *Allium cepa* assays to measure the genotoxicity. They did not impact cell viability when compared to controls and did not cause changes in the mitotic index at the quantities used. The presence of iron oxide nanoparticles did not affect seed germination [71].

## 6. Understanding the Mechanism of Synthesis of Trichogenic NPs

Fungal biosynthetic techniques can be grouped into intracellular and extracellular synthesis based on where nanoparticles are produced. Extracellular synthesis of nanoparticles, for example, is still being developed in terms of understanding the mechanisms of synthesis, simple downstream processing, and quick scale-up processing.

### 6.1. Bioactive Metabolites

The use of *T. longibrachiatum* for AgNP biosynthesis revealed the existence of proteins and their binding to AgNPs via carbonyl groups of amino acid residues and peptides, which might have contributed to their stability and prevention of agglomeration. Proteins on the surface of AgNPs function as capping agents [49]. There are 35 metabolites that have been discovered as significant variations between selenium nanoparticles biosynthesized from *T. harzianum* (TSNP) and standard SNPs, which included organic acids, sugars, amino acids, and carbohydrate metabolism intermediates. Among these, 27 are strong antifungal agents with higher levels in TSNPs than in SNPs. Many acids, sugars, and their derivatives are used in the coating of TSNPs, including heptonic acid, ferulate, fumaric acid, threonic acid, glucose, and mannitol. All of these organic compounds in aqueous formulations capped the selenium NPs and acted as stabilizers while also increasing the antagonistic capability of biosynthesized selenium NPs against pathogens [65]. While using *T. virens* HZA14 for AgNP biosynthesis revealed the interaction patterns of protein, carbohydrate, and heterocyclic compound molecules with AgNPs, the maximum yield was associated with gliotoxin [57]. Aromatic amino acids such as tyrosine and tryptophan were found in the FTIR spectra of *Trichoderma* spp. medium. The electrostatic attraction of negatively charged carboxylate groups by free amine groups, cysteine residues, or some proteins secreted by the fungus during the formation of AgNPs can bind to them. The release of extracellular protein molecules from fungus is responsible for the synthesis and stability of AgNPs [56]. The interaction of different functional groups of exometabolites with Ag was validated using FTIR for the production of green AgNPs and may be responsible for stabilization. *T. fusant* Fu21 inoculated with SM-containing chitin was used for GC–MS profiling to identify functional groups and compounds, which were discovered to be alkanes, dicarboxylic acid, aromatic ketones, amino acid, hetero-acyclic compounds, ketose sugar, sugar alcohol, aliphatic amines, polyol compounds, steroidal pheromones, and carbocyclic sugars [55]. Secondary metabolites released by *T. harzianum* operated as capping and reducing agents, providing consistency and contributing to biological activity determined by LC-MS/MS. The most common compounds included 1-benzoyl-3-[(S)-((2S,4R,8R)-8-ethylquinuclidin-2-yl] thiourea (6-methoxyquinolin-4-yl) methyl, puerarin, genistein, isotalatizidine, and ginsenoside [15]. The existence of numerous functional groups of biomolecules and capping protein, enclosing biosynthesized SiO_2_ NPs corresponding to carbonyl residues, alcohol, nitrile, acid chloride, alkene bands, and peptide bonds of the proteins involved, was verified by FTIR analysis [69]. The presence of functional groups of alkene, alkane, and alcohol in the FTIR test suggested that these may have participated in the SeNP production processes [66]. The presence of functional groups was verified by FTIR analysis of biosynthesized silica and copper nanoparticles using biocontrol agents, resulting in effective synthesis [63]. According to the FTIR analysis results, phenolic, proteins, amino acids, aldehydes, ketone, and other functional groups were involved in the reduction, capping, and stability of zinc oxide NPs [3].

### 6.2. Enzymes

While many microbial species can produce metal NPs, the mechanism of nanoparticle biosynthesis has not been identified as yet [77,78]. For its survival, *T. harzianum* produces enzymes and metabolites that are involved in the breakdown of silver nitrate into Ag^+^ ions and NO_3_, owing to the possible action of hydrolytic/nitrate reductase enzymes. Through the catalytic activity of extracellular fungal secondary metabolites, the toxic Ag^+^ ions are further converted to nontoxic (Ag^0^ = biosilver) metallic nanoparticles in this procedure. The bioreduction of silver is based on the presence of relevant functional groups in the extracellular filtrate of *T. harzianum*, which is more effective than other fungi and is nontoxic to humans [47,79]. FTIR and surface-enhanced resonance were used. Raman spectroscopy revealed a viable mechanism for the synthesis of silver nanoparticles in *T. asperellum* [44]. The procedure consisted of two critical steps: bioreduction of AgNO_3_ to produce AgNPs, followed by stabilization and/or encapsulation with a suitable capping agent. The cell’s defense mechanism for silver detoxification has been proposed as a biological process for AgNP production [80]. Extracellular enzyme secretion provides the advantage of obtaining large quantities in a relatively pure state, free of other cellular proteins associated with the organism, and can be easily processed by filtering the cells and isolating the enzyme for nanoparticle synthesis from the cell-free filtrate. *T. reesei* is thought to be the most effective extracellular enzyme producer compared to other filamentous fungi [39], and it has a long history of producing industrial enzymes [81]. The NADH co-enzyme, as well as NADH-dependent enzymes such as nitrate reductase, are present in *Trichoderma* genus strains and are essential in the formation of nanoparticles and capping that gives greater stability [39,82]. The determination of hydrolytic enzyme-specific activity revealed that AgNP-TS had a greater specific activity of NAGase and chitinase than AgNP-T. Concerning the filtrates, *T. harzianum* exposure to the cell wall of *S. sclerotiorum* enhanced the specific activity of the enzyme NAGase [43]. *T. harzianum* produces enzymes and metabolites involved in breaking strong ionic bonds between silver and nitrate ions for its own survival, perhaps by the action of hydrolytic/nitrate reductase enzymes. Extracellular fungal secondary metabolite enzymatic activity transforms dangerous Ag+ ions into nontoxic biosilver NPs [14].

## 7. Applications of *Trichoderma*-Mediated NPs in the Agri-Food Sector

Crop production invariably involves rampant use of chemical pesticides, insecticides, and herbicides that have been exhibiting bioaccumulation of toxic chemical residues potentially hazardous to plant ecosystems [40]. This problem has shifted the research foci to the identification of more effective alternatives such as nano-based agrochemicals. The intrusion of nano-interventions in the agri-food sector is being extended to extract the benefits showcased by nanotechnology applications in electronics, pharmaceutical, biomedical, paint, and cosmetics industries. However, the chemically synthesized nanoparticles have a high cost and substantial eco-toxicity (https://www.reportlinker.com/p06193716/Nanotechnology-Services-Global-Market-Report.html?utm_source=GNW, accessed on 20 February 2022). Their rampant use in open field conditions is anticipated to enhance the negative ecological and health hazards, besides the huge input costs. Therefore, green nanoparticles have been speculated to address these issues related to synthetic NPs. Various fungal species may create mycogenic nanoparticles that might stimulate growth and protect crops against diseases in some prospective agricultural uses through antioxidant, antimicrobial, and plant-stimulating properties. The primary benefits of the mycogenic NP synthesis protocol(s) will be the one-pot, cost-effective, and less environmentally corrosive features. Still, the widespread usage of myco-nanoparticles may lead to a few complications. For example, on multiple applications of these NPs, the nano-toxicity aspects will render the plant beneficial microbes vulnerable to the applied nanoparticles, which might compromise their viability as well as the biocontrol and plant growth promotion benefits. Farms and agricultural consumers will soon be able to utilize myconano-functional agrochemicals, pre-harvest and post-harvest crop protection agents, sensors used in genetic equipment, and crop protection components. Additionally, the fungus-derived nanoparticles can function as useful myco-nano-sorbents and would be an attractive way to perform heavy metal biosorption from contaminated wastewater. The employment of *Trichoderma* genus for the synthesis of metallic nanoparticles is ecologically advantageous, time-saving, and cost-effective. Furthermore, potential agrochemicals can be designed by amalgamating NPs and *Trichoderma* strains to produce more sustainable products. The agri-food applications of this technology outlined below will help you understand how the *Trichoderma* genus is used to make nanodevices and how these trichogenic NPs are being used in various applications.

### 7.1. Antifungal Activity

The effect of metal chitosan nanocomposites at 100 g mL^−1^ in combination with Cu-tolerant *Trichoderma longibrachiatum* strains on cotton seedling damping-off under greenhouse conditions was also investigated. In vitro, the bimetallic blends (BBs) and Cu-chitosan nanocomposite had the best antifungal effectiveness against both *R. solani* anastomosis groups. These findings suggested that BBs, the Cu chitosan nanocomposite, and BBs mixed with *Trichoderma* may inhibit *R. solani*-caused cotton seedling disease in vivo. *R. solani* was evaluated in a greenhouse with a *Trichoderma* strain and shown to have a synergistic inhibitory effect with BBs [83]. The biogenic synthesis of AgNPs was carried out by use of mycelial extracts of *T. harzianum* supplemented in aqueous silver nitrate (1 × 10^−3^ mol L^−1^). In a potato dextrose agar-based poison food assay, the use of concentrations ranging from 0.15 × 10^12^ and 0.31 × 10^12^ NPs/mL reduced the mycelial development and the generation of new sclerotia in *S. sclerotiorum* [4]. Antifungal application of AgNPs resulted in a considerable decrease in the number of forming colonies for several plant pathogenic fungi, with an efficiency of up to 90% against *Fusarium verticillioides, Fusarium moniliforme, Penicillium brevicompactum, Helminthosporium oryzae,* and *Pyricularia grisea* [49]. Because mycelial development was inhibited and no new sclerotia were formed, both AgNP-TS and AgNP-T nanoparticles showed promise for controlling *S. sclerotiorum*. The most effective inhibition of mycelial development was accomplished with AgNP-TS, which might be attributable to the nanoparticles’ reduced hydrodynamic diameter as well as a potential biomolecule effect from the nanoparticles’ capping [43]. With 20 g Ag/mL green nanoformulations for increased pathogen fungal mycelium breakdown activity, green Ag-NPs boost antifungal action to reduce the phytopathogen *Sclerotium rolfsii* producing stem rot in groundnut [55]. In vitro testing of AgNPs produced with *T. virens* HZA14 antifungal efficacy against *S. sclerotiorum* revealed that hyphal development, sclerotial formation, and myceliogenic germination of sclerotia were all inhibited by 100%, 93.8%, and 100%, respectively. The direct interaction between nanoparticles and fungal cells, including AgNPs’ contact, accumulation, lamellar fragment creation, and micropore or fissure development on fungal cell walls, was demonstrated using SEM/EDS technologies [57]. When compared to *T. asperellum* alone and carbendazim @0.1%, chitosan nanoparticles in combination with *T. asperellum* were found to be superior in suppressing the mycelial development of soil-borne pathogens such as *F. oxysporum*, *R. solani*, and *S. rolfsii* [68]. Copper and silica nanoparticles biosynthesized with biocontrol agents suppressed *P. hypolateritia* and *P. theae* growth, suggesting that this might be a unique way of managing diseases that affect tea plantations while also improving tea quality parameters. Various nanoformulations were prepared using suitably inert and eco-friendly carrier materials employing nano copper and nanosilica [63]. In the laboratory and greenhouse, *T. harizinum*-mediated ZnONPs were demonstrated, for the first time, to exhibit fungicidal activity against three soil–cotton pathogenic fungi (*Fusarium* sp., *R. solani*, and *Macrophomina phaseolina*) [3].

*Pseudomonas fluorescens* and *T.*
*viride*, two distinct biocontrol agents, were used to make microbial CuONPs, which were then evaluated using various analytical methods. In terms of in vitro antifungal efficacies, CuONPs synthesized from *T. viride* showed the highest percent growth inhibition compared to CuONPs generated from *P. fluorescens*. CuONPs produced from *T. viride*, on the other hand, demonstrated considerably stronger antifungal activity in vivo than the commonly used Bordeaux mixture [84]. Antifungal activity against three soil-borne pathogens was proven in vitro and in the greenhouse. In the three fungal infections, AgNPs greatly reduced hyphal growth. The ability of *T. harzianum* isolates to synthesize a wide spectrum of proteins and enzymes without the need for chemical reducers and stabilizers has been proven. Biosynthesized AgNPs have shown high potential in protecting cotton plants from the fungal invasion induced by damping-off [14]. *Trichoderma* species have a wide range of biotechnological uses, including acting as biofungicides to manage various plant diseases, biofertilizers to promote plant development, and synthesizing and bioremediating metal nanoparticles. Possible applications of mycogenic NPs generated by the *Trichoderma* genus are demonstrated in Figure 4.

### 7.2. Antibacterial Activity

The antibacterial activity of green nano-biosilver *T. harzianum* AgNPs was demonstrated against *S. aureus* and *K. aeruginosa*, as well as Gram-positive and Gram-negative bacteria, with the Gram-negative bacterium (*K. pneumoniae*) displaying better sensitivity. *T. atroviride*’s AgNPs were found to have antibacterial action against Gram-positive and Gram-negative clinical pathogens such as *E. coli*, *P. aeruginosa*, and *S. aureus* [46]. AgNPs had stronger antibacterial activity than AgNO_3_ and were similar to the positive control kanamycin. They also demonstrated DPPH scavenging action in a dose-dependent manner, with an IC50 of 45.6 g/mL. The cell-free filtrate, on the other hand, showed no inhibition against the pathogens, probably due to metabolites with the antibiotic property. As a result, the biosynthesized AgNPs can be employed as a natural antioxidant to prevent human cell damage and degenerative diseases by regulating antioxidants, pro-oxidants, and ROS levels [50]. The antibacterial activity of AgNP was evaluated against a variety of bacteria types. Gram-negative (*E. coli* and *P. aeruginosa*) bacteria had lower MBC and MIC values than Gram-positive (*S. aureus* and *E. faecalis*) bacteria, and this difference was ascribed to the bacterial cell wall structure. This suggests that Gram-negative bacteria have better antibacterial activity due to bacterial surface adsorption and oxidative stress induction; nevertheless, more research is required [56]. Gram-positive (*S. aureus* and *B. subtilis*) and Gram-negative (*E. coli* and *R. solanacearum*) pathogenic bacteria were tested, and the AgNPs synthesized using *T. harzianum* filtrate were reported to have significant antioxidant properties and antibacterial activity against both Gram-positive (*S. aureus* and *B. subtilis*) and Gram-negative *(E. coli* and *R. solanacearum*) bacteria, with higher activity against the Gram-negative bacteria [15]. CuNPs were effectively produced using *T. harzianum*, an agriculturally beneficial fungus, in a simple green and environmentally favorable way. CuNPs and *T. harzianum* fungus were shown to have antibacterial action against both Gram-positive and Gram-negative bacteria [62]. The antibacterial activity of the generated AgNPs against *Escherichia coli* was investigated [85].

### 7.3. Plant Growth Promotion

In greenhouse settings, *Trichoderma*-mediated AgNPs were evaluated against tomato wilt caused by *Fusarium* species. The treated plants with various doses of AgNPs showed a promoting effect on all the tested parameters in comparison with the control and *Trichoderma* formulation [86]. With the prolonged soaking time of silver nanoparticles solution, *T. harzianum*-synthesized AgNPs demonstrated an increase in the percentage of seed germination. AgNPs produced by *T. harzianum* had a positive effect on oilseed germination. As a result, mycogenic AgNPs have a biological assay for increasing seed viability in agriculture [87]. The biocontrol agent *T. harzianum* was used as a stabilizing agent in the green manufacture of biogenic iron oxide nanoparticles. The nanoparticles’ antifungal effectiveness against *S. sclerotiorum* (white mold) was tested in vitro. The impact of the NPs on seed germination was also examined. They were also able to promote the proliferation of *Trichoderma*, which inhibited the establishment of the pathogen *S. sclerotiorum* while having no effect on seed germination [71].

### 7.4. Trichoderma-Based Nanobioremediation

Fungi-based nanotechnology is quickly evolving as an effective technology to treat industrial wastewaters. *T. harzianum*-derived CdS-NPs were used to photocatalyze the breakdown of methylene blue in a photocatalytic cell [88]. The production of electron (e) and hole (h+) pairs, which function as powerful oxidants and reductants, caused the degradation. Adsorbed water on the CdS-NPs traps the hole (h^+^), resulting in the formation of a hydroxyl radical, while oxygen takes electrons from the conduction band, resulting in the formation of an anion radical. These radicals target the dye’s azo bond, causing it to degrade and produce CO_2_, H_2_O, and NH_4_^+^. Due to an effective biocatalyst that converted 4-nitrophenol to 4-aminophenol in the presence of NaBH_4_ and the capacity to inhibit pathogenic bacteria, biosynthesized gold nanoparticles present new hope for green bioremediation [67]. The test human pathogenic bacteria were strongly suppressed by biogenic AgNPs made from the fungus *T. viride* [48]. Because of their biocidal action against halotolerant planktonic sulfate-reducing bacteria, mycosynthesized AgNPs have become an appealing alternative for controlling microbially driven corrosion in the petroleum industry [53]. *T. harzianum*-biosynthesized SiO_2_ NPs were effective as a lead adsorbent from water during the bioremediation process. The content of lead in water and the muscles of Nile tilapia (*Oreochromis niloticus*) decreased. *O. niloticus*’ immune system, liver, and renal functions improved [69]. The *Trichoderma* genus can be employed as a possible metal biosorbent and as a powerful bioremediation agent. Some *Trichoderma* strains have high metal tolerance and bioaccumulation ability, making them a viable mycoremediation agent of heavy metal contamination in environmental safety. Figure 5 illustrates the pathways of tolerance to metals and nanoparticles (NPs) in *Trichoderma*.

### 7.5. Miscellaneous Advantages and Disadvantages of Trichogenic Nanoparticles

The biogenic AgNPs synthesized from the extracellular filtrate of *T. harzianum* have an additional benefit over physical and chemical approaches that need high pressure, energy, chemical precursors, and a high cost. This filamentous fungus has a larger capacity for binding Ag^+^ Ag^0^, making this procedure easier and less expensive for the large-scale manufacturing of NPs at the industrial level, making it cost-effective [47]. *T. viride’s* capacity to generate AgNPs is extremely promising for green, sustainable nanomaterial synthesis [48]. The nanoparticles demonstrated little size fluctuation as well as high physicochemical stability. As the beans were exposed to biogenic silver nanoparticles, there were no significant changes in germination and seedling development compared to the negative control [4]. *T. longibrachiatum* showed a possibility for the extracellular and reliable biosynthesis of silver nanoparticles. The AgNPs synthesized using this biosystem technique were generally stable up to two months after synthesis [49], and, even after six months of storage, nanocrystalline silver particles created by *T. asperellum* did not display significant aggregation [15,44]. The use of fungi in the biogenic synthesis of silver nanoparticles has several benefits, including the development of a capping agent from fungal biomolecules, which provides stability and can assist in biological activity. The cytotoxicity and genotoxicity of AgNP-TS and AgNP-T nanoparticles to V79, 3T3, and HaCat cell lines were both low [43]. When compared to normal pesticides, the use of nano-based formulations will be a preferable option for avoiding excess chemicals in soil [45]. *T. reesei* possesses simpler and less expensive culture conditions, as well as greater growth rates on both industrial and laboratory sizes, resulting in lower costs in large-scale manufacturing [39]. *T. atroviride*-based biosynthesized AgNPs are an economically effective and ecologically sustainable way of producing AgNPs [50]. The small-sized nanoparticles generated, their narrow size distribution, the stability attained without the use of chemicals as capping agents, and the low number of chemical reagents used are the key benefits of the described extracellular biosynthesis of AgNPs. Because of the negative zeta potential that might aid the attachment of physiologically active compounds, the generated AgNPs were appropriate carriers of such compounds [52,54]. The use of green AgNPs as an antifungal agent is seen as an environmentally friendly resource, an alternative to fungicides, and a cost-effective method [55]. The surface characteristics of the biogenic chitosan nanoparticles generated from *T. viride* were discovered to be positively charged and stable [68].

However, there are several drawbacks that must be resolved in order to properly employ fungus for biogenic synthesis. The engagement of the host organism for protein expression, immature synthesis techniques, and limited know-how on large-scale production technique(s) may possibly limit the benefits associated with fungus-based nanoparticle synthesis. The most critical factors that may affect the final product include knowing which fungus to employ, its growth parameters, the necessity for sterile settings, and the time it takes for the fungus to develop and finish the synthesis. Scaling-up can also create challenges, such as the need for research focusing on the mechanism associated with the development of capping layers and effective nucleation [68]. All these challenges associated with the green synthesis of various metal NPs through the use of different types of fungal biomass should be solved, and the feasibility of green synthesized nanoparticles for obtaining useful commercial applications in the agri-food industry should be identified.

## 8. Challenges

Mycogenic nanoparticles can be synthesized with a wide range of particle size distributions. There are issues with reproducibility as well as a lack of control over particle size, shape, and dispersion. As a result, in future efforts, the parameters of synthesis should be thoroughly investigated [89]. The development of innovative nanofungicides, nanofertilizers, and nanosorbents generated by fungi and the evaluation of these nanoformulations for the remediation of heavy metal-contaminated industrial wastewater are still in their early phases. Several problems must be overcome to scale up mycogenic nanoparticles and fulfill the demands of the real world. For example, a new nanosorbent created by diverse microorganisms must be economically advantageous and socially acceptable while also being able to meet various water quality regulations to ensure human health and environmental safety [90]. Scaling up the method is difficult since the commercial synthesis of nanoparticles still needs more study. No NP residue must be discharged into the environment as a result of the treatment method. For example, the environmental tracking of silver in the field is critical for evaluating its influence on the environment and human health. Furthermore, further study should be conducted to assess the hazardous effect of AgNPs before widespread manufacture and usage in agricultural applications [49]. The toxicity of NPs remains a key hurdle in transferring these materials from lab to industry for the time being, and this has to be addressed further.

## 9. Future Trends

The biogenic silver nanoparticles exhibited no deleterious effects on *T. harzianum* culture compared to the negative control. This shows that combining *T. harzianum* with nanoparticles would not affect the fungus’ development [4]. It is hoped that techniques involving the biogenic synthesis of metallic nanoparticles will be developed, taking into account not only the potential biological activity of the metal nanoparticles but also the biomolecules and organic compounds obtained from the organisms used in the synthesis, which compose the capping on the nanoparticles [43]. In order to have better control over the size and polydispersity of these nanoparticles, more research is needed to understand the specific chemical pathway leading to their production using biological means. More studies should be carried out on the creation of AgNPs-based compounds and their combination with fungicides [49]. Extracellular biosynthesized AgNPs might be used as carriers for different agrochemicals and natural bioactive molecules. There is a greater need for field trials of *Trichoderma*-based nanoparticles to understand nanoparticle–plant–microbe interactions and toxicity issues to develop safer and user-friendly agrochemicals.

## 10. Conclusions

Based on the literature review, we conclude on whether developing agrochemicals that employ *Trichoderma* strains to produce green NPs could be more sustainable agrochemicals. Given that the *Trichoderma* genus is one of the most often used fungi for the mycosynthesis of NPs, it is possible that it possesses tolerance mechanisms to these structures that might be used in combination to generate products that boost agricultural productivity while simultaneously treating plant infections in the field. The synthesis of NPs by beneficial fungi such as *Trichoderma* is novel, cost-effective, and eco-friendly compared to the synthesis by chemical and physical methods. However, more study is required to assess the *Trichoderma* and heavy metals interaction in a heterogeneous system under field circumstances. Additionally, to comprehend the potential advantages of nanoparticles produced by beneficial microbes in the agricultural sector, it is necessary to comprehend nanoparticle penetration and transport routes in plants. The shape and size of the nanoparticles have a significant impact on their reactivity, stability, and behavior. More effective agrochemical applications using composite materials are desirable because these can decrease potential economic losses and, in particular, environmental damage. For example, combining the antimicrobial impact of mycogenic nanoparticles with biofungicides for plant disease control may boost antifungal efficacy through synergistic interaction, allowing for a reduced fungicide dose and, as a result, avoiding the development of fungal pathogen resistance.

## Figures and Tables

**Figure 1 jof-08-00367-f001:**
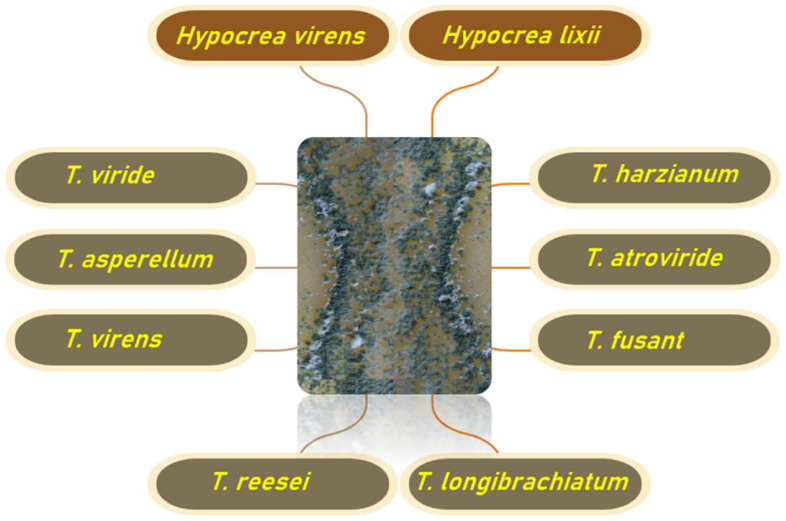
The top ten *Trichoderma* species used to produce safe metal nanoparticles through mycogenic synthesis.

**Figure 2 jof-08-00367-f002:**
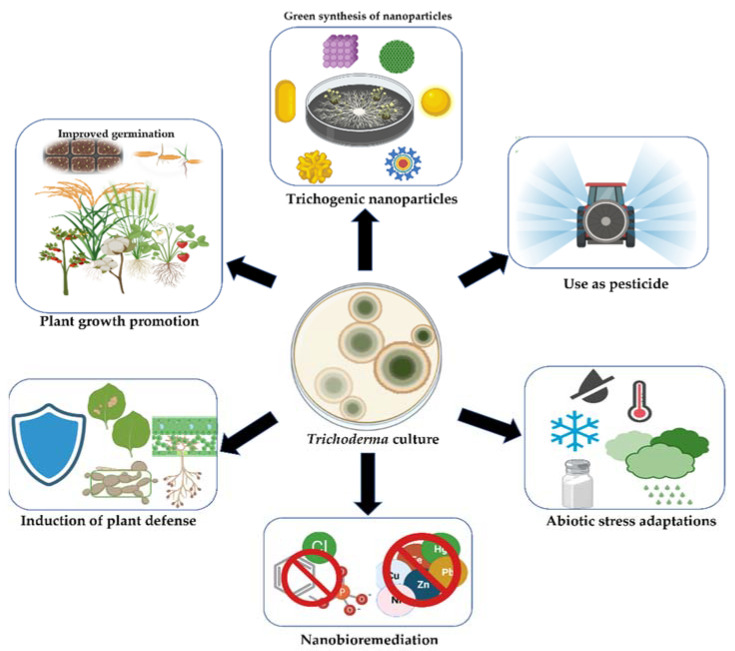
*Trichoderma* applications in the agricultural ecosystem. Red circle indicates stopping of the heavy metal contamination through active removal by nanomaterials. This figure was created with BioRender software.

**Figure 3 jof-08-00367-f003:**
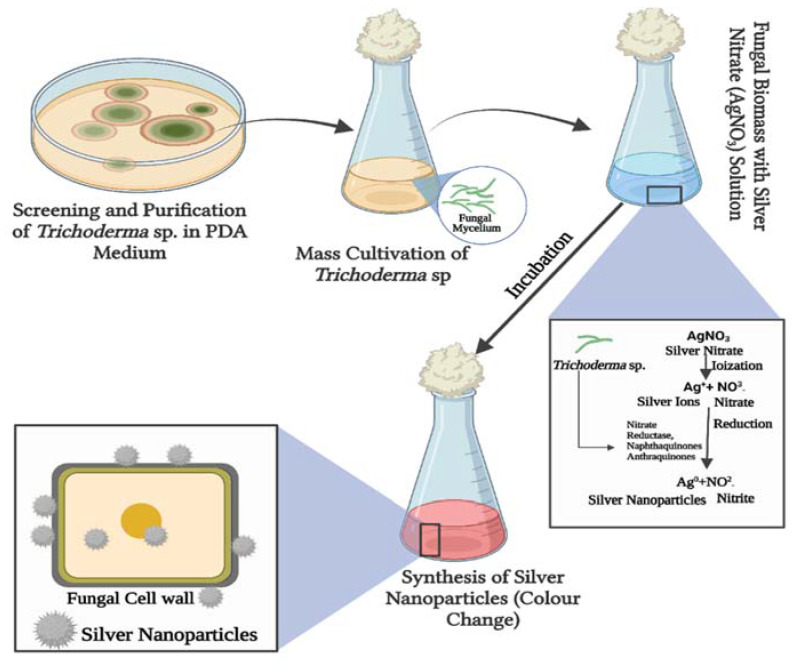
Schematic illustration of biosynthesis mechanism of silver nanoparticles (AgNPs) using *Trichoderma* species. The present figure was created by BioRender.com.

**Figure 4 jof-08-00367-f004:**
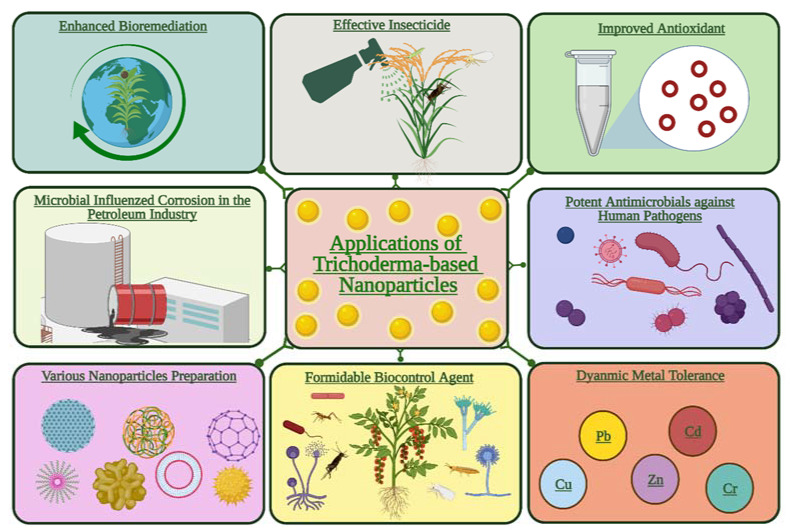
Various potential applications of *Trichoderma*-mediated nanoparticles in agroecosystems. The present figure was created by BioRender.com.

**Figure 5 jof-08-00367-f005:**
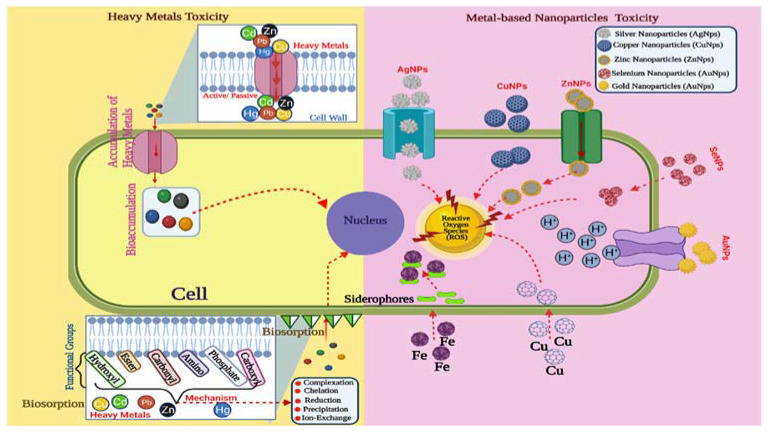
The adsorption and absorption mechanisms in the cell walls caused by the presence of functional groups, proteins, or compounds that serve as chelating agents, as well as the accumulation of these in the vacuoles, are illustrated schematically in the *Trichoderma* tolerance mechanisms to micro- and nano-metals. Mechanisms involving antioxidant enzyme activity that reduces the damage produced by reactive oxygen species may also be present. The arrows indicate movement of the metal ions or nanoparticles affecting specific organelle in the cell. The present figure was created by BioRender.com.

**Table 1 jof-08-00367-t001:** Trichoderma species employed for green synthesis metal nanoparticles.

*Trichoderma* Species	NPs	Size	Shape	Application	References
*T. asperellum*	AgNPs	13–18 nm	crystalline nature	Biomolecular detection	[44]
*T. reesei*	AgNPs	5–50 nm	variable morphology	Preparing many nanostructured materials and devices	[39]
*T. virens*	AgNPs	8–60 nm	round and uniform in shape	Crop protection	[45]
*T. harzianum*	AgNPs	10–51 nm	face centered cubic symmetry particles	Antioxidant properties and antibacterial activity	[46]
*T. harzianum*	AgNPs	10–20 nm	oval shaped, crystalline in nature	Mosquito control	[47]
*T. viride*	AgNPs	1–50 nm	globular particles	Antibacterial effect against human pathogenic bacteria	[48]
*T. harzianum*	AgNPs	20–30 nm	spherical	Control of *S. sclerotiorum*	[4]
*T. longibrachiatum*	AgNPs	5–25 nm	spherical	Control of many phytopathogenic fungi	[49]
*T. atroviride*	AgNPs	15–25 nm	anisotropic structural	Antioxidant and antibacterial against clinical pathogens	[50]
*T. harzianum*	AgNP-TS	57.02 ± 1.75 nm	different characteristics	Control of *S. sclerotiorum*	[43]
	AgNP-T	81.84 ± 0.67 nm			
*T. reesei*	AgNPs	1–4 nm 15–25 nm	crystal phase	carriers of biologically active molecules	[52]
*T. longibrachiatum* DSMZ 16517	AgNPs	5–11 ± 0·5 nm	spherical, triangular, and cuboid	Control of industrial microbes	[53]
*Trichoderma* sp.	AgNPs	14–25 nm	round	Antibacterial	[56]
*T. virens* HZA14	AgNPs	5–50 nm	spherical and oval with smooth surfaces	Control of *S. sclerotiorum*	[57]
*T. atroviride*	AgNPs	10–15 nm	spherical	Control of pathogenic bacteria and fungi	[54]
*T. fusant* Fu21	AgNPs	59.66 ± 4.18 nm	spherical	Control of *S. sclerotiorum*	[55]
*T. harzianum*	AgNPs	72 nm	cubic crystal structure	Antioxidant properties and antibacterial activity	[15]
*Trichoderma* spp. co-culture	ZnONPs	12–35 nm	crystal structure	Control of Bacterial Leaf Blight causative in rice	[59]
*T. harzianum* (SKCGW009)	ZnONPs	30.34 nm	spherical	Antibacterial activity enhanced roundworm growth	[58]
*T. harzianum*	ZnONPs	8–23 nm	hexagonal, spherical and rod	fungicidal action against three soil–cotton pathogenic fungi	[3]
*T. asperellum*	CuONPs	10–190 nm	spherical	development of anticancer nanotherapeutics	[60]
*T. harzianum*	CuONPs	~20 nm	spherical structure	Antibacterial activity	[62]
*T. harzianum*	AgNPs	5–18 nm	spherical	Control of plant pathogens	[61]
	CuONPs	38–77 nm	Dispersed and elongated fibers in shape		
	ZnONPs	27–40 nm in width	fan and bouquet structure	Control of microorganisms	
		134–200 nm in length			
*T. atroviride* and 2 other fungi	CuONPs	5–25 nm	spherical	Management of some tea plantation diseases	[63]
	SiO_2_NPs	12–22 nm			
*T. asperellum*	SeNPs	49.5–312.5 nm	hexagonal, near-spherical, and irregular	Control of *Sclerospora graminicola*, mildew disease causative in pearl	[64]
*T. harzianum* JF309	SeNPs	bigger than traditional SNP	irregular	antifungal	[65]
*Trichoderma* sp. WL-Go	SeNPs	An average of 147.1 nm	spherical and pseudo-spherical	ND	[66]
*T. viride*	AuNPs	20–30 nm	spherical	bioremediation	[67]
*T. viride*	Chitosan NPs	89.03 nm	nearly spherical	Control of soil borne pathogens	[68]
*T. harzianum* MF780864	SiO_2_NPs	89 nm	oval, rod, and cubical	Bioremediation	[69]
*T. harzianum*	iron oxide NPs	207 ± 2 nm (α-Fe_2_O_3_)	Spherical shape	Control of *S. sclerotiorum*	[71]

## Data Availability

Not applicable.

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
