# Peer review of "Trichoderma*: An Eco-Friendly Source of Nanomaterials for Sustainable Agroecosystems"

_jof, 2022, doi:10.3390/jof8040367_

Round 1
Reviewer 1 Report
jof-1640613-peer-review-v1
The review article is good addition in the body of knowledge and will have good impact for the researchers, my observations are as under for the improvement of the Review article
- The graphical abstract should be made if possible
- The contents shall be placed before the introduction section
- English needs to be improved from English native country
- In figures the names of the Trichoderma spp. should be italic
- Figure 111 cititation should be as mentioned athttps://help.biorender.com/en/articles/3619405-how-do-i-cite-biorender
- The application of nanoparticles has great potential in the field of Agriculture, it is advisable that the authors should add the cost-benefit ratio for the application. It will give more clear picture of its applicability in managing plant pathogens
- Advantages/merits and disadvantages/demerits of application of nanoparticles should be added in the manuscript
Author Response
Reviewer # 1:
The review article is good addition in the body of knowledge and will have good impact for the researchers, my observations are as under for the improvement of the Review article
- The graphical abstract should be made if possible
- The graphical abstract has been designed.
- The contents shall be placed before the introduction section.
- The journal author guidelines do not mention the requirement for incorporation of the TOC. Therefore, the same has not been prepared and incorporated.
- English needs to be improved from English native country.
- As suggested the manuscript thoroughly revised for use of English language.
- In figures the names of the Trichoderma spp. should be italic
- All scientific names have been rechecked and have been put in italic case.
- Figure 1 citation should be as mentioned at https://help.biorender.com/en/articles/3619405-how-do-i-cite-biorender
- Fig.1. has not been created with Biorender software rather it was designed in Microsoft powerpoint.
- The application of nanoparticles has great potential in the field of Agriculture, it is advisable that the authors should add the cost-benefit ratio for the application. It will give more clear picture of its applicability in managing plant pathogens.
- Based on the reviewer comments we have added the cost-benefit ratio for the applications of green synthesized NPs compared to synthetic NPs in agriculture.
- Advantages/merits and disadvantages/demerits of application of nanoparticles should be added in the manuscript
- Based on the reviewer suggestion, we have added the advantages and disadvantages of the Trichoderma-derived NPs. Please refer to section 7.5 for further clarity.
Reviewer 2 Report
This review article describes the utilisation of Trichoderma in nanoscience. The article covers all the new reports in nanoparticle production utilising Trichoderma and shows figures which help follow the text within the manuscript. In my opinion, the review article is suitable for publication at Journal of Fungi; however, I would suggest the authors to revise carefully as some in vitro/in vivo have not been italicised
Author Response
Reviewer # 2:
This review article describes the utilisation of Trichoderma in nanoscience. The article covers all the new reports in nanoparticle production utilising Trichoderma and shows figures which help follow the text within the manuscript. In my opinion, the review article is suitable for publication at Journal of Fungi; however,
- I would suggest the authors to revise carefully as some in vitro/in vivo have not been italicised
- As suggested by the reviewer, the words in vitro/in vivo have been modified to Italic case.
Reviewer 3 Report
This review focuses on nanoparticles produced by Trichoderma. Although the manuscript embraces an interesting overview of nanoparticles diversity, I think that a critical view of the topic is missing. Moreover, several points need substantial improvement, including references and figures. Please see below my detailed comments and suggestions.
- References are missing and should be provided to allow the reader to learn more about each concept. For example:
- Lines 101-105: references should be added to support these statements.
- Figure legends should be developed to better explain the content of the figures.
- Figures are globally very general, and some figures with more detailed information could increase the content of this review. A table gathering information about nanoparticles (type, size, species producing nanoparticles, functions of the nanoparticles, references…) could be added, for example.
- Repetitions happen regularly at the beginning of the manuscript.
- English language should be checked throughout the manuscript.
- For example, lines 272-273: “. among the four Amazon fungus isolates.” Words should be added to form a complete the sentence.
- Line 268: “sizes ranging from 59.66 4.18 nm“. Numbers should be corrected.
- In vitro should be in italic
- Line 53: I guess “enzymes” should be replaced by “metabolites”?
- Names of species should be in italic.
- All abbreviations should be explained. For example:
- Line 17: the abbreviation NP should be explained.
- lines 103-105: abbreviations should be explained.
- Line 717: what corresponds to the negative control?
Author Response
Reviewer # 3:
This review focuses on nanoparticles produced by Trichoderma. Although the manuscript embraces an interesting overview of nanoparticles diversity, I think that a critical view of the topic is missing. Moreover, several points need substantial improvement, including references and figures. Please see below my detailed comments and suggestions.
- References are missing and should be provided to allow the reader to learn more about each concept. For example:
- Lines 101-105: references should be added to support these statements.
- The relevant references have been incorporated as indicated by the reviewer.
- Figure legends should be developed to better explain the content of the figures.
- The figures have been developed as self-explanatory contents. We have tried to further improve the figure legends for better clarity.
- Figures are globally very general, and some figures with more detailed information could increase the content of this review. A table gathering information about nanoparticles (type, size, species producing nanoparticles, functions of the nanoparticles, references…) could be added, for example.
- The requisite information in the figures has been incorporated as per the suggestions of the reviewer. Table 1 has been incorporated illustrating the indicated aspect.
- Repetitions happen regularly at the beginning of the manuscript.
- The repetitive sentences have been modified or removed.
- English language should be checked throughout the manuscript. For example, lines 272-273: “. among the four Amazon fungus isolates.” Words should be added to form a complete the sentence.
- The manuscript has been revised for the use of English language. Also, the indicated incomplete sentence has been removed.
- Line 268: “sizes ranging from 59.66 4.18 nm“. Numbers should be corrected.
- The indicated change has been incorporated in the revised version.
- In vitro should be in italic
- This change has been applied in the revised manuscript.
- Line 53: I guess “enzymes” should be replaced by “metabolites”?
- We thank the learned reviewer for indicating this mistake. The word ‘enzyme’ has been replaced by ‘metabolites’ in the revised version of the manuscript.
- Names of species should be in italic.
- All the scientific names of the microorganisms have been rechecked and italicized.
- All abbreviations should be explained. For example:
- Line 17: the abbreviation NP should be explained.
- As indicated the abbreviation NP has been expanded and explained at the indicated section in the revised manuscript.
- lines 103-105: abbreviations should be explained.
- The abbreviations have been explained as indicated in the revised manuscript.
- Line 717: what corresponds to the negative control?
- Negative control corresponds to Trichoderma culture grown on potato dextrose media not supplemented with silver nanoparticles.
Reviewer 4 Report
The authors in their review article showed the possibility of the synthesis of nanoparticles of several metals with the use of Trichoderma strains and the potential agri-food application of thereof. The manuscript was written in a clear and concise way, but in my humble opinion, needs improvement prior to publication in the Journal of Fungi.
- The aim of the work as well as the novelty of the presented work should be specified in the introduction part. Moreover, the introduction part should be extended with some information about the Trichoderma genus, e.g. biology, biodiversity, phylogeny, just overall characteristics of the genus with the special emphasis on the 10 mentioned in the text species.
- 1 Caption – the microorganisms’ names should be written in italics. Similarly, in the reference list, the names of microorganisms are not written in italics.
- All of the abbreviations should be defined in the text, e.g. MAMP, DAMP, SAR, ISR, etc.
- "Table 1 presents many metal NPs produced by different Trichoderma species, including AgNPs,...". However, Table 1 is not present in the manuscript.
- I propose to draw a possible mechanism reaction of cations reduction to NPs by bioactive molecules to the 6.1. Subsection, as well as a reaction how the enzymes like hydrolytic/nitrate reductase are able to reduce Ag+ to AgNPs.
- Line 536 – please correct the numeric value
Author Response
Reviewer # 4:
- The aim of the work as well as the novelty of the presented work should be specified in the introduction part. Moreover, the introduction part should be extended with some information about the Trichoderma genus, e.g. biology, biodiversity, phylogeny, just overall characteristics of the genus with the special emphasis on the 10 mentioned in the text species.
- We thank the learned reviewer for suggesting these aspects related to Trichoderma. We would like to humbly submit that as this Review article is not related to biodiversity, and phylogeny of the fungus Trichoderma, the introduction section has been focused on the main objective of the article linked to the mycosynthesis of different types of metal nanoparticles from Trichoderma species for use in agri-food applications.
- 1 Caption – the microorganisms’ names should be written in italics. Similarly, in the reference list, the names of microorganisms are not written in italics.
- As suggested all the scientific names have been modified to italic case.
- All of the abbreviations should be defined in the text, e.g. MAMP, DAMP, SAR, ISR, etc.
- Based on the reviewer comments, all the abbreviations have been identified in the manuscript.
- "Table 1 presents many metal NPs produced by different Trichoderma species, including AgNPs,...". However, Table 1 is not present in the manuscript.
- The Table 1 has been inserted in the revised version of the manuscript.
- I propose to draw a possible mechanism reaction of cations reduction to NPs by bioactive molecules to the 6.1. Subsection, as well as a reaction how the enzymes like hydrolytic/nitrate reductase are able to reduce Ag+ to AgNPs.
- As suggested more information has been incorporated particularly description on the synthesis mechanisms through the fungal enzymes such as hydrolytic/nitrate reductase. The details have been summarized in Fig. 3. In addition we have designed a graphical abstract further summarizing the objectives of the article.
- Line 536 – please correct the numeric value
- The numeric value has been corrected as presented by the authors in their manuscript (Guilger et al 2017).
Round 2
Reviewer 3 Report
The manuscript has been improved. The authors took in account most of my suggestions, and I do not have further comments. Best regards.
Author Response
Thanks for your comments
Reviewer 4 Report
The manuscript has been carefully revised.
Author Response
Thanks for your comments